# Unlearning Virus Knowledge Toward Safe and Responsible Mutation Effect Predictions

## Abstract

Pre-trained deep protein models have become essential tools in fields such as biomedical research, enzyme engineering, and therapeutics due to their ability to predict and optimize protein properties effectively. However, the diverse and broad training data used to enhance the generalizability of these models may also inadvertently introduce ethical risks and pose biosafety concerns, such as the enhancement of harmful viral properties like transmissibility or drug resistance. To address this issue, we introduce a novel approach using knowledge unlearning to selectively remove virus-related knowledge while retaining other useful capabilities. We propose a learning scheme, PROEDIT, for editing a pre-trained protein language model toward safe and responsible mutation effect prediction. Extensive validation on open benchmarks demonstrates that PROEDIT significantly reduces the model's ability to enhance the properties of virus mutants without compromising its performance on non-virus proteins. As the first thorough exploration of safety issues in deep learning solutions for protein engineering, this study provides a foundational step toward ethical and responsible AI in biology.

## 1 Introduction

Pre-trained deep protein models are playing an increasingly important role in biological research (Narayanan et al., 2021; Pucci et al., 2022). By learning from massive amounts of existing protein data, these models uncover hidden relationships between protein sequences, structures, functions, and dynamics. Remarkable successes have been witnessed in diverse applications, such as enzyme design (Madani et al., 2023; Zhou et al., 2024b) and antibody screening (Wang et al., 2024a; He et al., 2024).

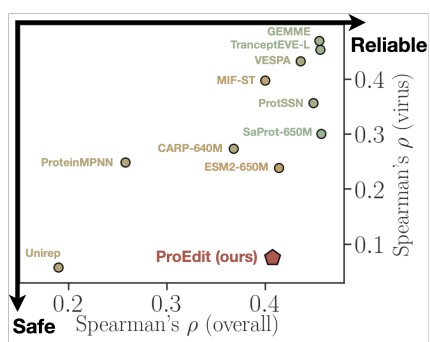

Figure 1: Positive Relationship of A model's overall performance vs Virus-related performance on mutation effect prediction. Data source: `https://proteingym.org/benchmarks`

Similar to natural language processing, pre-trained protein models often require training on billions of sequences with large-scale models to enhance expressivity and generalizability, achieving top performance across downstream tasks (Laine et al., 2019; Notin et al., 2022b; Lin et al., 2023b). This framework has been widely applied in solving problems in molecule design, where labels are usually scarce, expensive, or nonexistent. For instance, in enzyme engineering, *mutation effect prediction* (Notin et al., 2024) uses pre-trained models to score and rank the fitness of mutants relative to arbitrary wild-type proteins. Deep learning models guide proteins to modify toward enhanced functionalities such as activity, stability, and yield. Compared to previous rational design or simulation-based methods, they significantly improve mutation design success rates and reduce experimental costs by recommending better mutation strategies, while not relying on specific biological knowledge or experimental data (Lu et al., 2022; Li et al., 2023; Zhou et al., 2024a).

As is well known, a model's output is directly influenced by its training data. To improve expressivity and generalization, pre-trained models typically incorporate a large and diverse dataset to learn the parameters of the model. However, some of this knowledge may inevitably contain factual er-

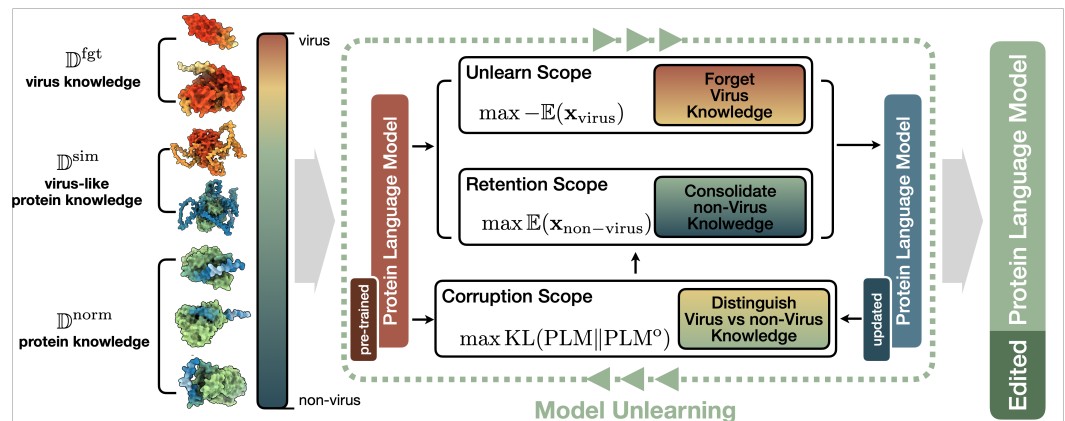

Figure 2: Illustration of the proposed PROEDIT. A pre-trained PLM is updated using three datasets to selectively forget virus-related knowledge while retaining non-virus information. To further enhance PROEDIT's ability to distinguish between virus and non-virus data, an additional dataset of virus-like proteins is specifically prepared to guide the optimization.

rors, biases, or even harmful information. These misleading elements can severely undermine the reliability and ethical integrity of the content generated by models pre-trained on such data. This issue is prevalent across many domains, such as natural language processing (Wang et al., 2024b) and computer vision (Golatkar et al., 2021).

The same concern has been raised in mutation effect prediction tasks. Figure 1 analyzes the performance of existing models on **ProteinGym** leaderboard (https://proteingym.org/benchmarks) for mutation effect prediction. It is evident that these powerful models show a strong positive correlation between reliability in modifying arbitrary enzymes and viruses. Designing tools capable of enhancing viral properties (*e.g.*, transmission, immune evasion, and drug resistance) and offering them to the public poses significant biosafety and ethical risks, such as disrupting ecological balance, triggering severe pandemics, and fostering biological weapons.

Therefore, it is urgent and important to develop corresponding techniques to edit protein models and allow the final models to retain their ability to effectively improve other enzymes while significantly reducing their capacity to enhance viruses, thereby mitigating ethical risks and enhancing the safety of research. While this issue has been preliminarily discussed in recent studies (Truong Jr & Bepler, 2023; Tan et al., 2023; Ouyang-Zhang et al., 2024; Liu et al., 2024), to the best of our knowledge, no solution has been developed to edit the pre-trained model and address this problem.

To this end, we employ the knowledge unlearning technique (Sinitsin et al., 2019; Wang et al., 2023) and propose PROEDIT, a learning scheme for safe and responsible protein language models (PLMs) for mutation effect prediction. We distinguish three types of data from the **UniRef** database: "virus", "non-virus", and "virus-like non-virus", and construct corresponding learning objectives. A pre-trained PLM is guided to retain its understanding of non-virus data within the retention scope while forgetting virus-related information within the unlearning scope (Figure 2). Notably, we introduce an additional corruption scope to ensure that the unlearned model retains the ability to understand virus-like non-virus data. Empirically, we validate that PROEDIT significantly reduces prediction performance on virus mutants across various virus assays while maintaining strong performance on non-virus mutants. In contrast, existing models either improve or degrade performance on both virus and non-virus assays simultaneously.

In summary, this work addresses mutation effect prediction–a core challenge in protein engineering–and presents the first detailed discussion of safety issues in deep learning solutions for this task. We propose a knowledge unlearning-based approach, which refines pre-trained models by distinguishing among three sets of training data and unlearning specific targets. This approach reduces ethical risks, specifically the ability of deep learning models to enhance the properties of viruses, while maintaining the model's overall performance in designing normal, non-harmful proteins. Comprehensive validation on multiple open benchmarks demonstrates the empirical significance of our proposed PROEDIT compared to existing solutions and ablation models in terms of effectiveness, consistency,

and efficiency. Although this work is an initial exploration, we believe that safety concerns in AI for biology are critically important and merit greater attention and discussion.

## 2 PRELIMINARY: MODEL PRE-TRAINING AND KNOWLEDGE UNLEARNING

### 2.1 PRE-TRAINED PROTEIN LANGUAGE MODEL FOR MUTATION EFFECT PREDICTION

PLMs are the mainstream approach for learning protein sequence representations, including both BERT-style and GPT-style pre-training schemes. The former learns to recover masked tokens in the input sequence, and the latter generates tokens autoregressively. For mutation effect prediction, we implement the BERT-style approach. During training, a BERT-style model applies random masks to the input sequence. The training objective is to find $\boldsymbol{\theta}$, the optimal parameters that minimize the difference between the prediction of the masked amino acids (AAs) and the corresponding ground truth, *i.e.*,

$$\arg\min_{\boldsymbol{\theta}} \mathbb{E}_{\boldsymbol{x}\sim\boldsymbol{X}} \mathbb{E}_M - \sum_{i\in M} \log \mathrm{P}(\boldsymbol{x}_i|\boldsymbol{x}_{/M}; \boldsymbol{\theta}). \tag{1}$$

The conditional probability $\mathrm{P}(\boldsymbol{x}_i|\boldsymbol{x}_{/M})$ of the $i$-th token $\boldsymbol{x}_i$ in the sequence is based on the unmasked part $\boldsymbol{x}_{/M}$. The model learns to interpret the interactions of AAs within the protein sequence.

The trained model $\mathrm{PLM}^{\mathrm{o}}$, obtained from (1), provides a summary matrix of the probability distribution for each AA in the sequence. This distribution has been shown to be effective for scoring mutation effects, especially when there is insufficient experimental data to support supervised learning. Given the AA probability distribution obtained from a pre-trained model $\mathrm{PLM}^{\mathrm{o}}$ for a wild-type protein, it can score relevant mutants of interest. Denote a $|\boldsymbol{F}|$-site mutant by a set of triplets $\boldsymbol{F} = \{(i, \boldsymbol{F}_i, \boldsymbol{w}_i)|i = 1, 2, \ldots, |\boldsymbol{F}|\}$, where $\boldsymbol{F}_i$ and $\boldsymbol{w}_i$ are the residue types of the $i$th AA after and before the point mutation, respectively. The fitness score of the mutant $\boldsymbol{F}$ is:

$$\mathrm{score}(\boldsymbol{F}) = \sum_{i=1}^{|\boldsymbol{F}|} \log \mathrm{P}(\boldsymbol{x}_i = \boldsymbol{F}_i|\boldsymbol{x}; \mathrm{PLM}^{\mathrm{o}}) - \log \mathrm{P}(\boldsymbol{x}_i = \boldsymbol{w}_i|\boldsymbol{x}; \mathrm{PLM}^{\mathrm{o}}). \tag{2}$$

The above zero-shot scoring function provides the *log-odds ratio* of mutants. Since most enzymes lack sufficient experimental labels to train a supervised learning model, this strategy is currently the most widely used scoring function in related research (Meier et al., 2021; Notin et al., 2024).

### 2.2 KNOWLEDGE UNLEARNING FOR PROTEIN LANGUAGE MODEL

Suppose an initial $\mathrm{PLM}^{\mathrm{o}}$ is trained on a collection of protein sequences with arbitrary properties $\mathbb{D}^{\mathrm{o}}$. For a new input $\boldsymbol{x}$, this well-trained model can provide the corresponding output $\boldsymbol{y} = \mathrm{PLM}^{\mathrm{o}}(\boldsymbol{x})$, regardless of whether $\boldsymbol{x}$ and the associated property to modify is desired or undesired.

However, as mentioned in the previous section, protein engineering tasks desire a safe and responsible model to provide reliable enhancement strategies for normal proteins (such as industrial enzymes), while being incapable of modifying harmful proteins (such as viruses). Formally, if there is a set of desired normal proteins $(\boldsymbol{x}^{\mathrm{norm}}, \boldsymbol{y}^{\mathrm{norm}}) \in \mathbb{D}^{\mathrm{norm}}$ and a set of undesired proteins $(\boldsymbol{x}^{\mathrm{fgt}}, \boldsymbol{y}^{\mathrm{fgt}}) \in \mathbb{D}^{\mathrm{fgt}}$, we aim to learn a modified $\mathrm{PLM}^{\mathrm{new}}$, which reduces the effectiveness of $\mathrm{PLM}^{\mathrm{o}}$ in understanding undesirable instances while maintaining its ability to infer desired instances, *i.e.*,

$$\boldsymbol{y}^{\mathrm{fgt}} \neq \mathrm{PLM}^{\mathrm{new}}(\boldsymbol{x}^{\mathrm{fgt}})$$
$$\text{and } \boldsymbol{y}^{\mathrm{norm}} = \mathrm{PLM}^{\mathrm{new}}(\boldsymbol{x}^{\mathrm{norm}}). \tag{3}$$

## 3 KNOWLEDGE UNLEARNING VIA MODEL RETRAINING

### 3.1 GENERAL OBJECTIVE

The overall goal of model unlearning in the context of mutation effect prediction is to reduce the model's ability to represent viruses while minimally affecting its representation capabilities for normal (non-virus) proteins. Methodologically, the aim is to update a pre-trained model $\mathrm{PLM}^{\mathrm{o}}$ into a new model $\mathrm{PLM}^{\mathrm{new}}$, *i.e.*, updating the parameters from $\boldsymbol{\theta}^{\mathrm{o}}$ to $\boldsymbol{\theta}$. To achieve this, we prepare three

training datasets corresponding to three optimization objectives: the unlearning scope, the retention scope, and the corruption scope. The unlearned model $\text{PLM}^{\text{new}}$ is expected to forget the knowledge in the unlearning scope while retaining the knowledge in the retention scope. Additionally, we define a corruption scope for virus-like proteins, on which we expect the unlearned model to maintain similar performance to the original model. Formally, we define the objective function as follows:

$$\arg\max_{\boldsymbol{\theta}} \quad \underbrace{-\mathbb{E}_{\boldsymbol{x}\sim\mathbb{D}^{\text{fgt}}}\log\text{P}_{\boldsymbol{\theta}}(\boldsymbol{x}|\boldsymbol{x}_M)}_{\text{Unlearn Scope}} + \underbrace{\mathbb{E}_{\boldsymbol{x}\sim\mathbb{D}^{\text{norm}}}\log\text{P}_{\boldsymbol{\theta}}(\boldsymbol{x}|\boldsymbol{x}_M)}_{\text{Retention Scope}}$$
$$+ \underbrace{\mathbb{E}_{\boldsymbol{x}\sim\mathbb{D}^{\text{sim}}}\text{KL}\left(\text{P}_{\boldsymbol{\theta}}(\boldsymbol{x}|\boldsymbol{x}_M)\|\text{P}_{\boldsymbol{\theta}^{\circ}}(\boldsymbol{x}|\boldsymbol{x}_M)\right)}_{\text{Corruption Scope}}. \tag{4}$$

**Unlearn Scope**  The first term, $-\mathbb{E}_{\boldsymbol{x}\sim\mathbb{D}^{\text{fgt}}}\log\text{P}_{\boldsymbol{\theta}}(\boldsymbol{x}|\boldsymbol{x}_M)$, measures the representation ability of $\text{PLM}^{\text{new}}$ in recovering masked tokens in viruses. The model's parameters are updated based on a virus dataset $\mathbb{D}^{\text{fgt}}$ to ensure the knowledge is forgotten. A model is considered to have effectively forgotten undesired virus knowledge if the recovery performance is poor, *i.e.*, if $-\mathbb{E}_{\boldsymbol{x}\sim\mathbb{D}^{\text{fgt}}}\log\text{P}_{\boldsymbol{\theta}}(\boldsymbol{x}|\boldsymbol{x}_M)$ is maximized.

**Retention Scope**  The second term, $\mathbb{E}_{\boldsymbol{x}\sim\mathbb{D}^{\text{norm}}}\log\text{P}_{\boldsymbol{\theta}}(\boldsymbol{x}|\boldsymbol{x}_M)$, measures the effectiveness of $\text{PLM}^{\text{new}}$ in recovering masked tokens in $\boldsymbol{x}^{\text{norm}}$, a set of non-virus normal proteins that are mutually exclusive to $\mathbb{D}^{\text{fgt}}$. A well-unlearned model is expected to minimize $\mathbb{E}_{\boldsymbol{x}\sim\mathbb{D}^{\text{norm}}}\log\text{P}_{\boldsymbol{\theta}}(\boldsymbol{x}|\boldsymbol{x}_M)$, indicating it retains knowledge relevant to normal proteins.

**Corruption Scope**  The third term, $\mathbb{E}_{\boldsymbol{x}\sim\mathbb{D}^{\text{sim}}}\text{KL}\left(\text{P}_{\boldsymbol{\theta}}(\boldsymbol{x}|\boldsymbol{x}_M)\|\text{P}_{\boldsymbol{\theta}^{\circ}}(\boldsymbol{x}|\boldsymbol{x}_M)\right)$, focuses on a challenging subset of virus-like proteins $\mathbb{D}^{\text{sim}} \subset \mathbb{D}^{\text{norm}}$. As an augmentation, this term requires the unlearned model to minimize the difference (measured by the KL divergence) between $\text{PLM}^{\text{new}}$ and the original model $\text{PLM}^{\text{o}}$, ensuring that forgetting viral knowledge does not disrupt the knowledge of virus-like normal proteins.

## 3.2 Training Scheme

**Data Preparation**  To implement the untraining scheme, we divide the **UniRef50** dataset [1] into three sets. The first two sets, $\mathbb{D}^{\text{fgt}}$ and $\mathbb{D}^{\text{norm}}$, are directly split from the processed **UniRef50** dataset based on their annotated Taxon IDs. Specifically, $\mathbb{D}^{\text{fgt}}$ includes proteins whose Taxon IDs indicate a biological lineage of viruses [2]. The remaining proteins form $\mathbb{D}^{\text{norm}}$. These two datasets, after processing, contain $65,511,306$ and $564,268$ sequences, respectively. The statistics of these sequences are detailed in Appendix A.1. For the virus-like proteins $\mathbb{D}^{\text{sim}}$, we extract them from $\mathbb{D}^{\text{norm}}$ using a retrieval module. Specifically, for each virus protein $\boldsymbol{x}^{\text{fgt}} \in \mathbb{D}^{\text{fgt}}$, we pair the $k$-nearest proteins from $\mathbb{D}^{\text{norm}}$ based on the cosine similarity of their ESM-2 (650M) sequence embeddings. After preparing all three datasets, we conduct a random split on each of them, resulting in corresponding training, validation, and test sets with a ratio of 8:1:1.

**Model Optimization**  The trainable parameters $\boldsymbol{\theta}$ of $\text{PLM}^{\text{new}}$ are updated iteratively based on (4). To enhance training stability, we adopt an alternating micro-batch training strategy. Samples from the same batch originate exclusively from one of the datasets $\mathbb{D}^{\text{fgt}}$, $\mathbb{D}^{\text{norm}}$, or $\mathbb{D}^{\text{sim}}$, as defined in (5):

$$\mathcal{L}_{\text{batch}} = \begin{cases} -\frac{1}{|\mathcal{B}|}\sum_{\boldsymbol{x}\in\mathcal{B}}\log\text{P}_{\boldsymbol{\theta}}(\boldsymbol{x}|\boldsymbol{x}_M), & \text{if } \mathcal{B} \subset \mathbb{D}^{\text{norm}} \\ \frac{1}{|\mathcal{B}|}\sum_{\boldsymbol{x}\in\mathcal{B}}\log\text{P}_{\boldsymbol{\theta}}(\boldsymbol{x}|\boldsymbol{x}_M), & \text{if } \mathcal{B} \subset \mathbb{D}^{\text{fgt}} \\ \frac{1}{|\mathcal{B}|}\sum_{\boldsymbol{x}\in\mathcal{B}}\text{KL}\left(\text{P}_{\boldsymbol{\theta}}(\boldsymbol{x}|\boldsymbol{x}_M)\,\|\,\text{P}_{\boldsymbol{\theta}^{\circ}}(\boldsymbol{x}|\boldsymbol{x}_M)\right), & \text{if } \mathcal{B} \subset \mathbb{D}^{\text{sim}}. \end{cases} \tag{5}$$

This approach ensures that the model focuses on a single objective at a time, thereby improving convergence and preventing interference between different learning objectives. The stopping criteria include four key considerations concerning perplexity and Spearman's $\rho$:

1. Perplexity of sampled data from $\mathbb{D}^{\text{norm}}$ (smaller is better);

---

[1] https://www.uniprot.org/help/downloads

[2] The biological lineage of a Taxon ID can be obtained from NCBI at https://www.ncbi.nlm.nih.gov/

2. Perplexity of sampled data from $\mathbb{D}^{\text{fgt}}$ (larger is better);

3. Spearman's $\rho$ for assays of virus from ProteinGym (smaller is better);

4. Spearman's $\rho$ for assays of normal proteins from ProteinGym (larger is better).

We randomly sample $10\%$ of instances as validation set. At the end of each epoch, we compute these metrics and the training will be stopped if any of the four metrics decreases for ten consecutive epochs.

## 3.3 ALTERNATIVE UNLEARNING METHODS

In addition to the optimization method we proposed above, strategies from other frameworks can also be adopted to unlearn PLMs. Below we brief four alternative strategies employed in unlearning LLMs. Their performance in unlearning PLMs will be compared in the subsequent section.

**Gradient Ascent** The first method uses gradient ascent (Tian et al., 2024) to forget learned knowledge. The learning objective remains consistent with the pre-training stage. When untraining a pre-trained MLM, this method trains on $\mathbb{D}^{\text{norm}}$ and updates the model parameters by performing gradient ascent, *i.e.*, the opposite of descent, on (1).

**Model Corruption with Random Labels** The second approach is to fine-tune the model using randomly generated labels (Golatkar et al., 2020). Intuitively, by associating the data to be forgotten with random or incorrect labels, the model is expected to unlearn the associations it had previously made. In our case, this method trains using both $\mathbb{D}^{\text{fgt}}$ and $\mathbb{D}^{\text{norm}}$. The ground truth labels from $\mathbb{D}^{\text{fgt}}$ are randomly replaced with uniformly sampled tokens from the vocabulary, while the labels from $\mathbb{D}^{\text{norm}}$ remain unchanged and are used to train with gradient descent.

**Joint Gradient Ascent and Descent** The third hybrid method leverages gradient ascent to forget undesired information and gradient descent to retain useful knowledge (Yao et al., 2023). By alternating between the two, the model aims to forget specific information while retaining as much overall performance as possible. We apply this strategy to untrain a PLM on both $\mathbb{D}^{\text{fgt}}$ and $\mathbb{D}^{\text{norm}}$, using (1) as the training objective. Gradient ascent is applied to $\mathbb{D}^{\text{fgt}}$, while gradient descent is applied to $\mathbb{D}^{\text{norm}}$. It can be considered as a variant of PROEDIT that omits the corruption scope.

**Gradient Ascent with KL Constraint** The last strategy uses KL-divergence (Yao et al., 2023) to constrain the model's outputs, ensuring they do not stray too far from the original knowledge during the unlearning process. This approach helps balance the unlearning task by maintaining a good trade-off between forgetting and retaining information. The model is updated by performing gradient ascent on $\mathbb{D}^{\text{fgt}}$ and using KL divergence on $\mathbb{D}^{\text{norm}}$. Notably, the KL divergence on $\mathbb{D}^{\text{norm}}$ ensures that the model's outputs remain consistent with those of the original model. The key difference between this method and the second method (joint gradient ascent and descent) is that the former uses KL divergence to maintain output consistency, whereas the latter employs the MLM pre-training objective to prevent forgetting knowledge of normal (non-virus) proteins.

## 4 EXPERIMENTS

### 4.1 EXPERIMENTAL PROTOCOL

**Setup** We compare the performance of PROEDIT and baseline methods on different benchmark datasets. We use pre-trained ESM-2 (150M) and ESM-2 (650M) (Lin et al., 2023b) as the two base models and apply PROEDIT for editing. To improve training efficiency, we limit each epoch to a maximum of $2,000$ batches, with each batch containing 4 samples. All trainable parameters are updated using the ADAM (Kingma & Ba, 2015) optimizer with a learning rate of $1 \times 10^{-5}$. The MLM pre-training objective remains consistent with that of ESM-2, as defined in (1). Specifically, $15\%$ of the tokens in each sequence are selected for masking: $80\%$ are replaced with the "`[MASK]`" token, $10\%$ are substituted with a random token, and the remaining $10\%$ are left unchanged. For the four alternative unlearning methods introduced in Section 3.3, we adopt the same hyper-parameter configurations as PROEDIT. All implementations are done using `PyTorch` (version 1.7.0), and the

Table 1: Statistics Summary on the three benchmarks.

| Name | Type | # Assays (virus) | # Assays (non-virus) | # Mutations | # Train Samples | # Test Samples |
|------|------|------------------|----------------------|-------------|-----------------|----------------|
| **ProteinGym** | Zero-Shot | 30 | 187 | 2,465,767 | - | 2,465,767 |
| **AAV** | Supervised | 1 | 0 | 82,583 | 1,170 | 81,413 |
| **GB1** | Supervised | 0 | 1 | 8,733 | 5,089 | 3,644 |

experiments are run on an NVIDIA® RTX 4090 GPU with 24GB VRAM, mounted on a server with Ubuntu 22.04 LTS operating system. All the details to reproduce our results are included in the submission, and the code will be made publicly available upon acceptance.

**Benchmark Datasets**   We conduct a comprehensive evaluation of PROEDIT's editing capabilities, including both zero-shot and fine-tuning performance. The zero-shot prediction is evaluated on 217 deep mutational scanning (DMS) assays from **ProteinGym** (Notin et al., 2024), while fine-tuning is assessed on two supervised learning tasks: a viral dataset (**AAV**, Adeno-associated virus) and a non-viral dataset (**GB1**, binding domain of protein G, from Streptococcal bacteria) (Dallago et al., 2021). The statistics of these three benchmarks are summarized in Table 1. The training and evaluation procedures for these three tasks are as follows:

1. **ProteinGym** includes 30 viral assays and 187 non-viral assays. For each model, including PROEDIT, we score mutants using (2) and calculate the Spearman's $\rho$ correlation between the predicted and experimental mutational scores. In evaluating model performance, we aim for a high (closer to 1) Spearman's $\rho$ on non-viral assays and a low Spearman's $\rho$ on viral assays.

2. **AAV** is used for the first supervised learning task, which is a viral protein dataset originating from the FLIP benchmark. We use the "one-vs-rest" split, which consists of $1,170$ single-order mutations for training and $81,413$ high-order mutations for testing. Model performance is evaluated using Spearman's $\rho$, where a good model should have a lower Spearman's $\rho$.

3. **GB1**, a binding protein from Streptococcal bacteria, is used for the second supervised learning task. It is another dataset from the FLIP benchmark. We use the "low-vs-high" split from the FLIP benchmark, which includes $5,089$ low-fitness mutations for training and $3,644$ high-fitness mutations for testing. Similar to AAV, model performance is evaluated using Spearman's $\rho$. In this case, a good model should have a higher Spearman's $\rho$.

For both supervised learning tasks, **GB1** and **AAV**, we added a regression MLP head to the model, which was fine-tuned on the respective training sets. The fine-tuning process used the following hyperparameters: the ADAMW (Kingma & Ba, 2015; Loshchilov et al., 2017) optimizer with a learning rate of 0.0005, a weight decay of 0.01, a batch size of 16, and a dropout rate of 0.1 for the output layer.

## 4.2 RESULTS ANALYSIS

We evaluate the performance of the proposed PROEDIT from three dimensions: effectiveness, consistency, and efficiency. The performance comparison of PROEDIT and baseline methods on mutation effect prediction tasks is reported in Table 2. We use ESM-2 (650M) and ESM-2 (150M) as our two base models. We also compare several other pre-trained PLM models, including MIF-ST Yang et al. (2023), CARP Yang et al. (2024), and ESM-1v Meier et al. (2021). Additionally, we include a vanilla Transformer Vaswani et al. (2017) for the two supervised learning tasks. The architecture of vanilla Transformer is the same as ESM-2 and the only difference is that the parameters of the vanilla Transformer are randomly initialized before training. The initialization configuration is according to ESM-2 and the random seed is 42.

**Effectiveness.**   From the results reported in Table 2, it can be observed that PROEDIT significantly reduces the performance of the base model on **ProteinGym**-virus and **AAV**, while maintaining performance on **ProteinGym**-non-virus and **GB1**. This indicates that PROEDIT effectively assists the pre-trained base model in unlearning viral knowledge in both zero-shot and fine-tuning tasks. Specifically, for PROEDIT (650M), compared to ESM-2 (650M), the Spearman's $\rho$ correlation on **ProteinGym**-virus dropped from 0.24 to 0.08, retaining only 30% of the original performance. On

Table 2: Spearman's $\rho$ correlation of mutation effect prediction by different methods.

| Model | version | zero-shot prediction | | fine-tuning | |
| | | ProteinGym (virus) ↓ | ProteinGym (non-virus) ↑ | AAV ↓ | GB1 ↑ |
|---|---|---|---|---|---|
| Transformer | vanilla | - | - | 0.30 | 0.08 |
| MIF-ST | - | 0.40 | 0.43 | - | 0.22 |
| CARP | 640M | 0.27 | 0.37 | 0.43 | 0.48 |
| ESM-1v | - | 0.28 | 0.44 | 0.37 | **0.27** |
| ESM-2 | 150M | 0.13 | 0.45 | 0.08 | 0.15 |
| ESM-2 | 650M | 0.24 | **0.48** | 0.35 | 0.17 |
| PROEDIT | 150M | 0.08 | 0.43 | -0.16 | 0.13 |
| PROEDIT | 650M | **0.07** | 0.47 | **-0.18** | 0.24 |

† The top two are highlighted by **First** and Second.

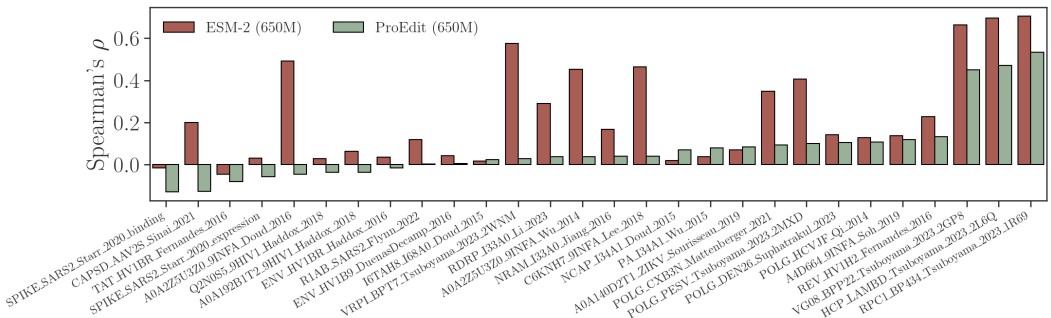

Figure 3: Individual prediction of assays in **ProteinGym** (virus) by PROEDIT and ESM-2.

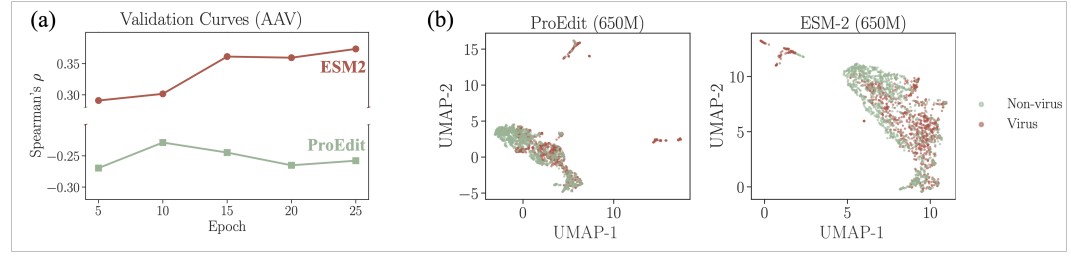

Figure 4: (a) Validation curves and (b) UMAP embedding of PROEDIT (650M) and ESM-2 (650M).

the viral protein **AAV**, PROEDIT (650M) dropped to $-0.18$ and PROEDIT (150M) dropped to $-0.16$, which are significantly lower than the original scores at $0.34$ and $0.08$, respectively. The detailed prediction performance on individual assays in **ProteinGym** (virus) is visualized in Figure 3, where PROEDIT greatly reduces the Spearman's $\rho$ on the majority of the assays in comparison to the performance of the base model (ESM-2-650M), with specific numbers provided in Appendix A.2. Additionally, comparing the results in Figures 4(b)-(c) shows that PROEDIT successfully scrambled viral information after unlearning. Here, we randomly selected $1,000$ virus and $1,000$ non-viral protein sequences and extracted hidden representations using both PROEDIT (650M) and ESM-2 (650M), followed by dimension reduction with UMAP (McInnes et al., 2018). It is evident that, compared to ESM-2, the representations of virus and non-viral proteins encoded by PROEDIT are more indistinguishable, further indicating the successful unlearning of viral proteins by PROEDIT. Similarly, Figure 4(a) displays the validation curves for PROEDIT (650M) and ESM-2 (650M) trained on **AAV** (virus) in a supervised learning setup. It shows clearly that the performance of PROEDIT remains constantly at a low level during training on this virus dataset, demonstrating the effectiveness of the unlearning process.

**Consistency** When the model forgets viral knowledge, it should minimize any negative impact on general protein knowledge. Specifically, the performance reduction in scoring non-viral proteins be-

Table 3: Zero-shot prediction performance on **ProteinGym** by PROEDIT and alternative unlearning methods.

| Base Model | Edit Method | ProteinGym (Virus) | | ProteinGym (non-Virus) | |
|---|---|---|---|---|---|
| | | Perplexity ↑ | Spearman's $\rho$ ↓ | Perplexity ↓ | Spearman's $\rho$ ↑ |
| ESM-2-650M | No Edit | 1.27e+00 | 0.238 | 1.18e+00 | 0.475 |
| | Gradient Ascent | 2.22e+15 | -0.123 | 3.92e+15 | -0.110 |
| | Joint Gradient Ascent and Descent | 1.18e+14 | 0.073 | 1.46e+03 | 0.309 |
| | Random Labels | 3.58e+00 | 0.116 | 1.81e+00 | 0.415 |
| | Gradient Ascent with KL Constraint | 1.94e+14 | 0.065 | 1.24e+02 | 0.383 |
| | PROEDIT | 9.06e+07 | 0.073 | 1.66e+00 | 0.470 |
| ESM-2-150M | No Edit | 1.34e+00 | 0.135 | 1.23e+00 | 0.449 |
| | Gradient Ascent | 1.11e+12 | 0.136 | 1.15e+12 | 0.122 |
| | Joint Gradient Ascent and Descent | 5.70e+10 | -0.087 | 2.19e+03 | 0.232 |
| | Random Labels | 2.93e+00 | 0.094 | 1.36e+00 | 0.433 |
| | Gradient Ascent with KL Constraint | 7.89e+10 | -0.065 | 1.04e+02 | 0.383 |
| | PROEDIT | 9.05e+06 | 0.083 | 1.75e+00 | 0.430 |

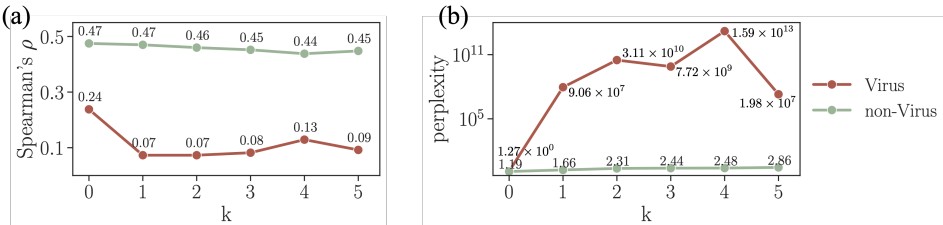

Figure 5: Performance of PROEDIT at different values of $k$ in the retrieval module on **ProteinGym**.

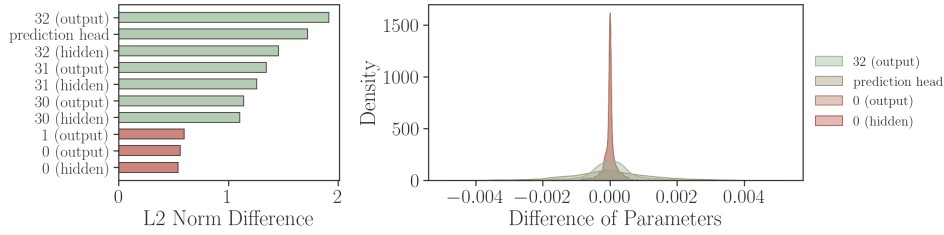

Figure 6: Parameter differences between ESM-2 and PROEDIT (650M).

fore and after unlearning should be minimized. This performance can be observed by analyzing the scores in Table 2 for **ProteinGym** (non-virus) and **GB1**. On the **ProteinGym** (non-virus), the score of PROEDIT (650M) decreased by only 0.01 compared to ESM-2 (650M), and PROEDIT (150M) decreased by merely 0.02 compared to ESM-2 (150M). Furthermore, on **GB1**, comparing the same parameter versions of ESM-2 and PROEDIT reveals that the unlearned model achieves better performance on this non-viral protein than the base model, surpassing all other baseline models. This is evident that the unlearning method is capable of retaining general protein knowledge when removing undesired virus knowledge.

**Efficiency**  We show that editing pre-trained PLMs with PROEDIT would not introduce significant additional computational costs. In our experiments, training PROEDIT costs 7.27 GPU hours for updating parameters in ESM-2 (650M), and 5.25 GPU hours for updating parameters in ESM-2 (150M).

### 4.3 ADDITIONAL INVESTIGATIONS

**Alternative Unlearning Methods**  Table 3 compares the zero-shot prediction performance of four alternative unlearning strategies mentioned in Section 3.3 on **ProteinGym**. Although all five unlearning methods (including ours) can assist a pre-trained PLM in forgetting harmful knowledge

about viruses, the alternative methods significantly reduce performance on non-viral proteins at the same time. In contrast, our PROEDIT is more practically meaningful. It is capable of maintaining both effectiveness and consistency, thereby enhancing model safety while preserving reliability on general protein tasks.

**Construction of the Virus-Like Dataset**  Figure 5 explores the impact of different choices on $k$ in the retrieval module when constructing $\mathbb{D}^{\text{sim}}$ on the results. Panels (a) and (b) report the changes in Spearman's $\rho$ and perplexity as $k$ varies, with $k = 0$ indicating the scores without editing. As $k$ increases gradually from 1 to 5, the model's performance on non-viral proteins slowly declines. A possible explanation is that increasing $k$ would directly enlarge $\mathbb{D}^{\text{sim}}$, causing some normal proteins that are not very similar to viruses to be included in the corruption scope, thus interfering with the model parameter updates. Additionally, in **ProteinGym** (virus), increasing $k$ does not help the model forget viral information more effectively. Therefore, in practice, we set $k = 1$ to perform an effective and efficient unlearning scheme.

**Change of Parameters Before and After Unlearning**  Figure 6 further investigates the changes in learnable parameters of the Transformer layers before and after editing to provide additional insights on the overall impact of the unlearning module. Taking PROEDIT (650M) and ESM-2 (650M) as examples, Figure 6(a) displays the ten network layers with the largest and smallest differences in L2 norm. The specific L2 norm of all parameters of the Transformer layers are provided in Appendix A.3. Overall, among the 33 Transformer layers, parameters of layers closer to the output show greater changes, while layers closer to the input exhibit smaller changes. This indicates that the model's forgetting primarily occurs in the last few layers of the Transformer. Figure 6(b) illustrates the parameter changes of the layers with the largest and smallest variations. Despite the differing magnitudes of change, both exhibit a symmetric bell-shaped distribution.

## 5 RELATED WORK

**Pre-trained Protein Language Models**  Analogous to NLP models, PLMs typically treat AAs as tokens and use Transformer-based layers (Vaswani et al., 2017) to analyze co-evolutionary information from millions to billions of protein sequences and summarize vector representations for sequences. Pre-trained PLMs can be categorized into three types. The most common are encoder-only models, which follow the BERT framework (Devlin et al., 2018) and train the model to recover randomly masked AA types (Meier et al., 2021; Rives et al., 2021; Elnaggar et al., 2021; Tan et al., 2024b). Decoder-only models, in comparison, are trained to optimize next-token prediction, which is frequently used for sequence design (Ferruz et al., 2022; Notin et al., 2022a; Madani et al., 2023). Other models adopt hybrid encoder-decoder architectures to learn outputs that are sufficiently similar to the input sequences (Du et al., 2022; Elnaggar et al., 2023; Heinzinger et al., 2023).

**Mutation Effect Prediction**  When applying models to enhance protein properties and functionalities, some studies adopt a "pre-training then fine-tuning" approach to enhance the model's understanding of a particular protein assay, such as using existing experimental data for supervised learning (Li et al., 2023; Zhou et al., 2024c; Tan et al., 2024a) or incorporating homologous sequences during training (Rao et al., 2021; Notin et al., 2022b). However, due to the lack of publicly available mutation effect data for most proteins and the variability among assays, the mainstream approach remains zero-shot methods for mutation effect prediction. Considering the pivotal role of structure in determining a protein's function, many recent methods integrate geometric deep learning methods (Lu et al., 2022; Tan et al., 2023; Zhou et al., 2024a; Tan et al., 2024c) or extract structure tokens (Su et al., 2023; Li et al., 2024a) to enhance the local interaction of spatially connected AAs. With the introduction of large-scale deep mutational scanning benchmark datasets like ProteinGym (Notin et al., 2024), an increasing number of models have been developed and extensively validated on a wide range of protein assays to demonstrate their effectiveness and generalizability.

**Knowledge Unlearning and AI Safety**  As emerging models grow larger and training data becomes more diverse, increasing attention is being directed toward developing approximate unlearning algorithms, such as data-reversed training (Chundawat et al., 2023) and optimization-based unlearning (Guo et al., 2020; Neel et al., 2021). In the NLP field, particularly with LLMs, AI safety has caught increasing attention. Knowledge unlearning, in this context, trains models to reject sensitive

responses (Yu et al., 2023; Yao et al., 2023; Tian et al., 2024; Li et al., 2024b). Similar discussions have emerged in other fields such as Computer Vision (Kim & Woo, 2022; Lin et al., 2023a; Tarun et al., 2023). However, in AI for biology, particularly in protein engineering, the safety and responsibility of developed deep learning models remain under-explored.

## 6 Conclusion and Discussion

This study addresses a critical safety concern in mutation effect prediction, a core task in protein engineering, by proposing a novel knowledge unlearning-based framework, PROEDIT. Our approach enables pre-trained PLMs to selectively forget virus-related information while preserving their capacity to predict and design non-viral proteins. Through empirical validation on multiple benchmarks, we demonstrated that PROEDIT effectively reduces the risk of enhancing viral properties without compromising the performance of beneficial proteins. This contributes to the growing need for ethical and responsible AI in scientific applications, particularly in biosafety-sensitive domains like protein engineering.

The rapid development of deep learning techniques in recent years has led to an increasing number of powerful solutions to biological challenges. The growing attention to these advancements has significantly driven improvements in the prediction and generation performance of biological entities, such as drug discovery, enzyme engineering, and protein design. However, alongside these technological advancements, we emphasize that ensuring their ethical and responsible use is equally crucial. We hope this work inspires more researchers to explore safety concerns in AI-driven protein engineering and to extend the unlearning framework to other safety-critical applications. By developing models that excel in predictive power while also addressing potential risks, the scientific community can promote safer and more responsible advancements in AI-driven biological research.

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

# A APPENDIX

## A.1 DATASET STATISTICS

This section presents statistics of the viral sequences and non-viral sequences in **UniRef50**, including the number of sequences, amino acid distribution, sequence lengths, and more.

**Virus Sequences** There are a total of **564,268** sequences. The length distribution and amino acid distribution of them are shown in Table 7 and Table 8, respectively.

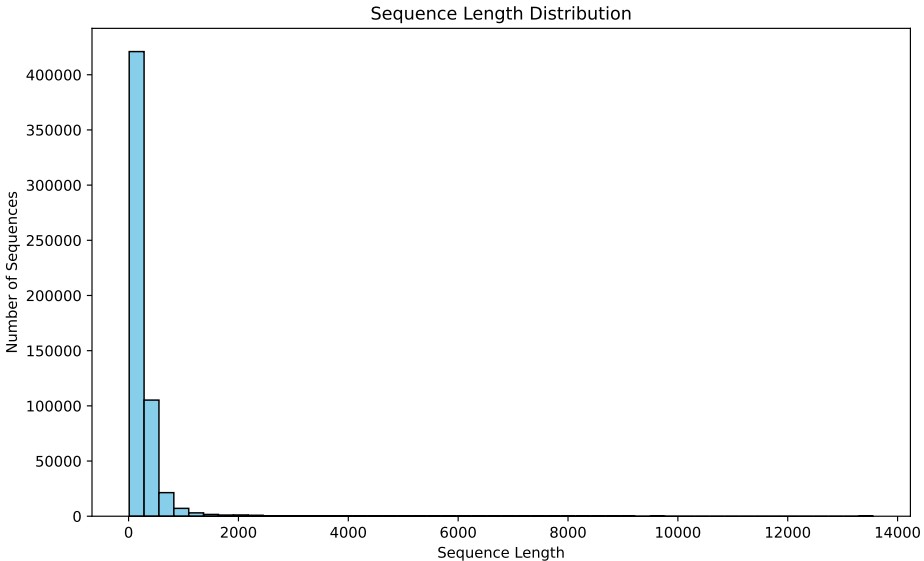

Figure 7: Length distribution of virus sequences in **UniRef50**.

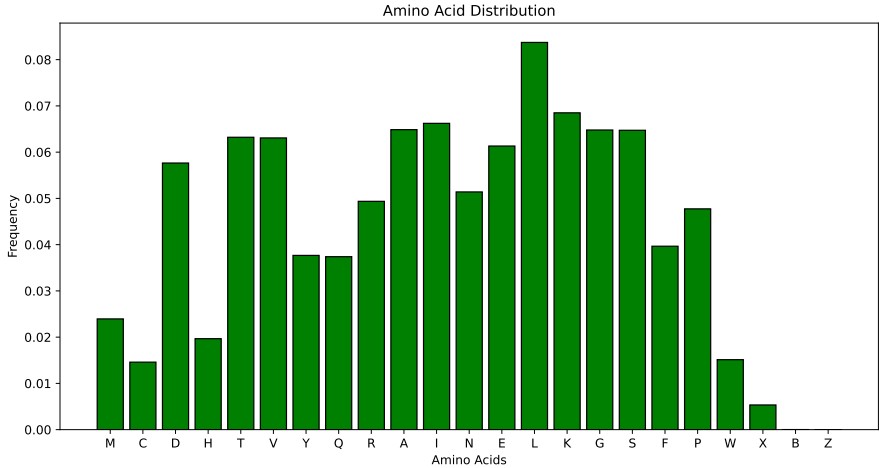

Figure 8: Amino acid distribution of virus sequences in **UniRef50**.

**Non-virus Sequences** There are a total of **65,511,306** sequences. The length distribution and amino acid distribution of them are shown in Table 9 and Table 10, respectively.

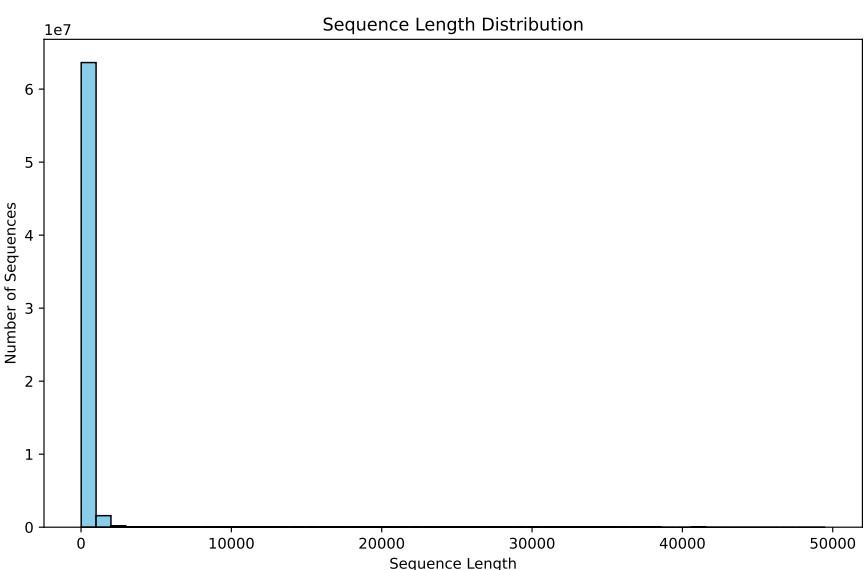

Figure 9: Length distribution of non-virus sequences in **UniRef50**.

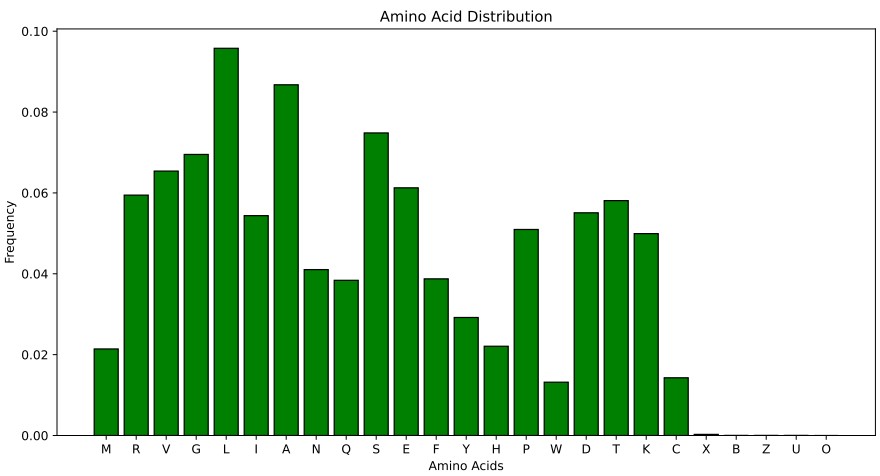

Figure 10: Amino acid distribution of non-virus sequences in **UniRef50**.

**Virus-like non-virus Sequences**    There are a total of **564,268** sequences (equal to virus sequences since $k = 1$) . The length distribution and amino acid distribution of them are shown in Table 11 and Table 12, respectively.

A.2    INDIVIDUAL PREDICTION OF ASSAYS IN PROTEINGYM (VIRUS).

Table 4 shows the detailed prediction performance on individual assays in **ProteinGym** (virus).

A.3    PARAMETER DIFFERENCES OF ESM-2 (650M) AND PROEDIT (650M)

Table 5 shows the detailed prediction performance on individual assays in **ProteinGym** (virus).

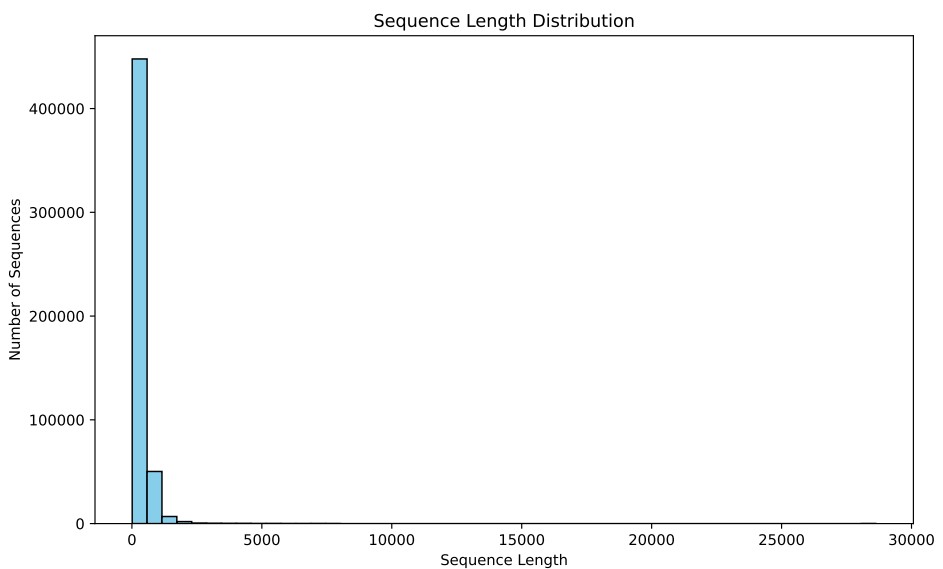

Figure 11: Length distribution of virus-like non-virus sequences in **UniRef50**.

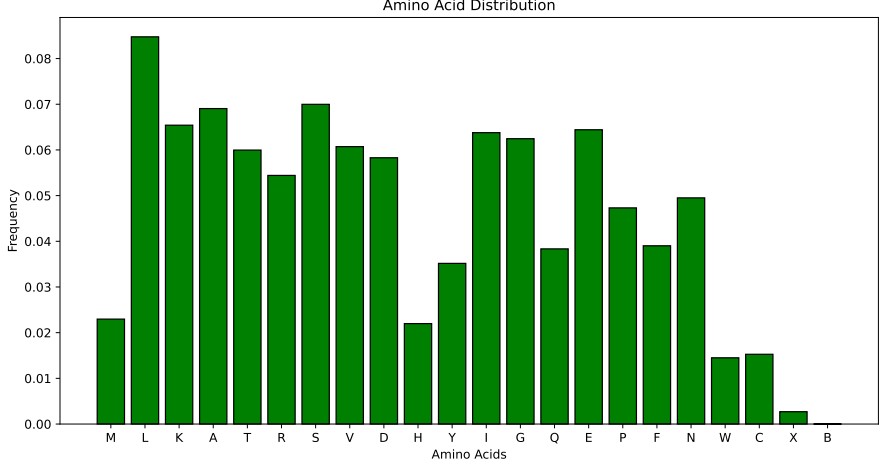

Figure 12: Amino acid distribution of virus-like non-virus sequences in **UniRef50**.

Table 4: Spearman's $\rho$ correlation of mutation effect prediction by PROEDIT in ProteinGym (virus).

| DMS_ID | PROEDIT (650M) | ESM-2 (650M) |
|---|---|---|
| A0A140D2T1_ZIKV_Sourisseau_2019 | 0.084 | 0.071 |
| A0A192B1T2_9HIV1_Haddox_2018 | -0.037 | 0.064 |
| A0A2Z5U3Z0_9INFA_Doud_2016 | -0.047 | 0.492 |
| A0A2Z5U3Z0_9INFA_Wu_2014 | 0.039 | 0.452 |
| A4D664_9INFA_Soh_2019 | 0.118 | 0.137 |
| C6KNH7_9INFA_Lee_2018 | 0.041 | 0.464 |
| CAPSD_AAV2S_Sinai_2021 | -0.127 | 0.200 |
| ENV_HV1B9_DuenasDecamp_2016 | 0.006 | 0.042 |
| ENV_HV1BR_Haddox_2016 | -0.015 | 0.036 |
| HCP_LAMBD_Tsuboyama_2023_2L6Q | 0.471 | 0.695 |
| I6TAH8_I68A0_Doud_2015 | 0.023 | 0.017 |
| NCAP_I34A1_Doud_2015 | 0.070 | 0.020 |
| NRAM_I33A0_Jiang_2016 | 0.040 | 0.166 |
| PA_I34A1_Wu_2015 | 0.079 | 0.038 |
| POLG_CXB3N_Mattenberger_2021 | 0.092 | 0.349 |
| POLG_DEN26_Suphatrakul_2023 | 0.105 | 0.143 |
| POLG_HCVJF_Qi_2014 | 0.106 | 0.127 |
| POLG_PESV_Tsuboyama_2023_2MXD | 0.100 | 0.406 |
| Q2N0S5_9HIV1_Haddox_2018 | -0.037 | 0.028 |
| R1AB_SARS2_Flynn_2022 | 0.003 | 0.118 |
| RDRP_I33A0_Li_2023 | 0.038 | 0.290 |
| REV_HV1H2_Fernandes_2016 | 0.133 | 0.227 |
| RPC1_BP434_Tsuboyama_2023_1R69 | 0.534 | 0.705 |
| SPIKE_SARS2_Starr_2020_binding | -0.129 | -0.015 |
| SPIKE_SARS2_Starr_2020_expression | -0.057 | 0.030 |
| TAT_HV1BR_Fernandes_2016 | -0.081 | -0.045 |
| VG08_BPP22_Tsuboyama_2023_2GP8 | 0.450 | 0.662 |
| VRPI_BPT7_Tsuboyama_2023_2WNM | 0.029 | 0.576 |

Table 5: Parameter differences' L2 norm between the hidden and output of transformer layers of ESM-2 (650M) and PROEDIT (650M).

| Parameter | L2 Norm | Parameter | L2 Norm |
|---|---|---|---|
| 32 (output) | 1.916 | 20 (hidden) | 0.934 |
| prediction head | 1.724 | 21 (output) | 0.933 |
| 32 (hidden) | 1.458 | 19 (hidden) | 0.932 |
| 31 (output) | 1.348 | 7 (hidden) | 0.932 |
| 31 (hidden) | 1.260 | 11 (output) | 0.932 |
| 30 (output) | 1.140 | 10 (output) | 0.931 |
| 30 (hidden) | 1.105 | 9 (output) | 0.928 |
| 29 (output) | 1.058 | 17 (hidden) | 0.928 |
| 28 (output) | 1.048 | 20 (output) | 0.928 |
| 29 (hidden) | 1.030 | 19 (output) | 0.926 |
| 28 (hidden) | 1.005 | 18 (hidden) | 0.925 |
| 10 (hidden) | 0.993 | 6 (hidden) | 0.923 |
| 9 (hidden) | 0.984 | 13 (output) | 0.918 |
| 11 (hidden) | 0.982 | 12 (output) | 0.918 |
| 27 (hidden) | 0.981 | 17 (output) | 0.914 |
| 27 (output) | 0.976 | 18 (output) | 0.914 |
| 26 (hidden) | 0.965 | 16 (output) | 0.912 |
| 26 (output) | 0.960 | 15 (output) | 0.911 |
| 25 (hidden) | 0.956 | 14 (output) | 0.910 |
| 12 (hidden) | 0.955 | 7 (output) | 0.907 |
| 24 (hidden) | 0.948 | 8 (output) | 0.904 |
| 23 (hidden) | 0.947 | 5 (hidden) | 0.904 |
| 25 (output) | 0.945 | 6 (output) | 0.898 |
| 13 (hidden) | 0.945 | 5 (output) | 0.888 |
| 8 (hidden) | 0.945 | 4 (hidden) | 0.863 |
| 22 (hidden) | 0.943 | 4 (output) | 0.855 |
| 24 (output) | 0.943 | 3 (output) | 0.748 |
| 23 (output) | 0.941 | 3 (hidden) | 0.745 |
| 22 (output) | 0.940 | 2 (output) | 0.720 |
| 15 (hidden) | 0.938 | 2 (hidden) | 0.719 |
| 14 (hidden) | 0.938 | 1 (hidden) | 0.603 |
| 21 (hidden) | 0.937 | 1 (output) | 0.596 |
| 16 (hidden) | 0.935 | 0 (output) | 0.560 |
| | | 0 (hidden) | 0.544 |