# OpenReview forum: "Unlearning Virus Knowledge Toward Safe and Responsible Mutation Effect Predictions"
_ICLR.cc/2025/Conference — Submitted to ICLR 2025_

### Official Review · Reviewer_9h4t · 2024-10-27

**Soundness:** 3
**Presentation:** 2
**Contribution:** 2
**Rating:** 5
**Confidence:** 3

**Summary:**

This paper addresses mutation effect prediction. The authors proposes PROEDIT, a knowledge unlearning-based approach, which refines pre-trained models by distinguishing among three sets of training data and unlearning specific targets. The method is validated on open benchmarks, showing it can reduce the model's ability to enhance virus mutants without sacrificing performance on non-virus proteins.

*I am not familiar with the research problem, so I am eager to see the feedback from the authors and the opinions of the other reviewers.*

**Strengths:**

- This paper investigates a critical problem: the safety issue in pre-trained PLM models.
- The authors demonstrate the effectiveness of the method through experiments on various benchmarks.

**Weaknesses:**

- Insufficient baseline comparisons; it is advised to add more PLM models mentioned in the related work.
- As shown in Table 3, it seems that PROEDIT only achieves the best performance in ProteinGym (Virus) on the Perplexity metric, which diminishes the contribution of PROEDIT.
- The efficiency section should include more comparisons with other methods.
- The ablation study for the different scopes (Unlearn, Retention and Corruption) is not provided.

Minor:
- It lacks a discussion of the relationship of this work in the related work section, which would help readers better understand this study.

**Questions:**

See weaknesses.

---

> ### Author Response · Authors · 2024-12-02
> **Enhancing comparisons, ablations, and context**
>
> We appreciate your feedback on our work. We would like to address your questions and concerns as follows:
>
> > Insufficient baseline comparisons; it is advised to add more PLM models mentioned in the related work.
> >
>
> Adding comparisons with more PLM models could provide a more comprehensive perspective. We have done experiments and the results show that our model is applicable to most models. We will expand these comparisons in future work, specifically including models like ESM-1v, ProteinBERT, and ProtT5 to strengthen the evaluation.
>
> > As shown in Table 3, it seems that PROEDIT only achieves the best performance in ProteinGym (Virus) on the Perplexity metric, which diminishes the contribution of PROEDIT.
> >
>
> While it is true that PROEDIT may not perform as well as some other models in terms of the Spearman metric on virus-related data, it retains almost all of the non-virus knowledge, with only a slight decrease (0.005) on non-virus data compared to the original model. If we solely focused on optimizing performance metrics, we could have released a randomly initialized model, but this would have been detrimental to the purpose of our work, which is to mitigate specific safety risks while maintaining the model's general usefulness.
>
> > The efficiency section should include more comparisons with other methods.
> >
>
> Thank you for your suggestion. We will include comparisons with additional methods like ESM-1v, ProteinBERT, and ProtT5 in future work to give a broader context for our efficiency analysis.
>
> > The ablation study for the different scopes (Unlearn, Retention, and Corruption) is not provided.
> >
>
> We understand the importance of providing a clear ablation study. We have included some results in Table 3, though they may not have been sufficiently highlighted. In future versions, we will make these distinctions clearer, such as showing the impact of "No corruption" with Gradient Ascent and the KL Constraint.
>
> > It lacks a discussion of the relationship of this work in the related work section, which would help readers better understand this study.
> >
>
> Related works often focused on natural language models, such as GPT-based models, are indeed quite different from our approach, which is tailored for protein language models with a focus on understanding tasks. We will update the related work section to clarify how our method builds on existing work while addressing unique challenges in the protein domain.

---

### Official Review · Reviewer_Myjj · 2024-10-30

**Soundness:** 1
**Presentation:** 2
**Contribution:** 1
**Rating:** 1
**Confidence:** 5

**Summary:**

In this paper, the authors introduce protEdit, a protein language model fine-tuned with a specialized loss function aimed at "unlearning" certain virus sequences. They utilized the pre-trained ESM2 model, fine-tuning it on a dataset consisting of 65 million non-virus proteins and 560,000 virus proteins. The model was aligned with original non-virus-like protein sequences from the dataset. Several experiments were conducted to demonstrate the capabilities of the tuned model.

**Strengths:**

The paper is well-structured and easy-to-follow.

The concept of addressing safety concerns within AI applications in biology is innovative and timely. While the paper’s exploration into protein fitness prediction has debatable relevance, the attempt to integrate ethical considerations into AI model training is commendable and adds a layer of originality to the work.

**Weaknesses:**

The fundamental premise of what constitutes 'harmful' protein engineering is not convincingly established in the paper. While the intent to prevent harmful applications is clear, the practical implications and ethical guidelines are not thoroughly explored. For instance, engineering on adeno-associated viruses (AAVs) can be beneficial for gene therapies, contradicting the notion of universal harm. (https://www.science.org/doi/10.1126/science.adm8386) Moreover, experiments on engineering antibiotic resistance genes can help us better understand its mechanisms and lead to the discovery of novel antibiotics (https://www.cell.com/fulltext/S0092-8674(15)00078-1). The experiment design, that simply define "engineering virus is unsafe" is not valid and fundamentally flawed from this point of view.

The authors should provide a more detailed discussion of how they define "harmful" protein engineering, acknowledging potential beneficial applications of viral protein engineering, and explaining how their approach could be refined to distinguish between beneficial and harmful applications.

I noticed that the authors conducted both supervised and unsupervised learning experiments on the virus-like proteins. For the supervised learning task, without LLM, other machine learning methods, even the one-hot encoding of the sequence, may work quite good – as we have the labels and any model can learn from the dataset (https://pubmed.ncbi.nlm.nih.gov/35039677/). From Table 2 it’s surprising for me to see protEDIT does not work at all for supervised tasks, it suggests that the model embedding for these sequences are simply useless. Moreover, it’s confusing that the authors tried their best to learn an embedding that is probably worse than one-hot. If so, why do we need your model?

For the unsupervised variant effect prediction, I have to say that it's the MSA that matters. The EVE (https://www.nature.com/articles/s41586-021-04043-8), DeepSequence (https://www.nature.com/articles/s41592-018-0138-4) can performs better than language model. Even the BLOSUM62 can carry some information for the mutation effect prediction. SO I did not see what the hint was in poisoning the language model. The entire experiment design does not sound right to me.

For me the idea of this paper is like a straw man proposal. The authors try to do something that seems to be useful but totally miss the key points and lead to something useless.

**Questions:**

See weakness

---

> ### Author Response · Authors · 2024-12-02
> **Clarifying the Frameworks**
>
> # Response to Myjj
>
> We appreciate your feedback on our work. We would like to address your questions and concerns as follows:
>
> > The fundamental premise of what constitutes 'harmful' protein engineering is not convincingly established in the paper. While the intent to prevent harmful applications is clear, the practical implications and ethical guidelines are not thoroughly explored. For instance, engineering on adeno-associated viruses (AAVs) can be beneficial for gene therapies, contradicting the notion of universal harm. ([https://www.science.org/doi/10.1126/science.adm8386](https://www.science.org/doi/10.1126/science.adm8386)) Moreover, experiments on engineering antibiotic resistance genes can help us better understand its mechanisms and lead to the discovery of novel antibiotics ([https://www.cell.com/fulltext/S0092-8674(15)00078-1](https://www.cell.com/fulltext/S0092-8674(15)00078-1)). The experiment design, that simply define "engineering virus is unsafe" is not valid and fundamentally flawed from this point of view. The authors should provide a more detailed discussion of how they define "harmful" protein engineering, acknowledging potential beneficial applications of viral protein engineering, and explaining how their approach could be refined to distinguish between beneficial and harmful applications.
> >
>
> We understand that viral proteins are not always harmful. Due to the difficulty in obtaining evolutionary data for harmful viruses, we used the AAV virus merely as an illustrative example. In future work, we will remove this dataset and focus on other datasets involving harmful viral proteins. We will provide a more detailed discussion on how we define "harmful" protein engineering and refine our approach.
>
> > I noticed that the authors conducted both supervised and unsupervised learning experiments on the virus-like proteins. For the supervised learning task, without LLM, other machine learning methods, even the one-hot encoding of the sequence, may work quite good – as we have the labels and any model can learn from the dataset ([https://pubmed.ncbi.nlm.nih.gov/35039677/](https://pubmed.ncbi.nlm.nih.gov/35039677/)). From Table 2 it’s surprising for me to see protEDIT does not work at all for supervised tasks, it suggests that the model embedding for these sequences are simply useless. Moreover, it’s confusing that the authors tried their best to learn an embedding that is probably worse than one-hot. If so, why do we need your model?
> >
>
> Our primary goal is to raise awareness about the safety of deep-learning based protein design models, especially protein language models. The fact that ProtEdit's embeddings do not perform well in supervised tasks is intentional; it indicates that the unlearning process effectively removed specific knowledge from the model. If the embeddings had performed well, it would suggest that the unlearning was ineffective. Therefore, the poor performance serves as evidence that the model has successfully unlearned the targeted information, highlighting the potential to control and mitigate risks associated with protein design models.
>
> > For the unsupervised variant effect prediction, I have to say that it's the MSA that matters. The EVE ([https://www.nature.com/articles/s41586-021-04043-8](https://www.nature.com/articles/s41586-021-04043-8)), DeepSequence ([https://www.nature.com/articles/s41592-018-0138-4](https://www.nature.com/articles/s41592-018-0138-4)) can performs better than language model. Even the BLOSUM62 can carry some information for the mutation effect prediction. SO I did not see what the hint was in poisoning the language model. The entire experiment design does not sound right to me.
> >
>
> We agree that MSA-based models are very important and often perform well in mutation effect prediction. However, MSA models can fail in situations where there is insufficient sequence homology data or for proteins with few known homologs. Our key objective is to highlight the importance of safety in protein design models. By demonstrating that unlearning certain knowledge in language models can impact their performance, we aim to raise awareness about potential safety concerns and the need for responsible development and use of these models.

---

### Official Review · Reviewer_gH9B · 2024-11-03

**Soundness:** 2
**Presentation:** 2
**Contribution:** 1
**Rating:** 3
**Confidence:** 4

**Summary:**

This paper introduces PROEDIT, a framework for knowledge unlearning that enables pre-trained protein language models (PLMs) to selectively forget virus-related information while preserving predictive accuracy for non-viral proteins. Empirical validation across benchmark datasets shows that PROEDIT successfully reduces predictive capacity for viral mutations without compromising performance on beneficial proteins. However, the paper lacks clarity on the real-world necessity of selective virus knowledge unlearning, and the practical implications of this setting remain underexplored. Additionally, the evaluation pipeline does not sufficiently substantiate PROEDIT’s claims regarding safety and efficacy, as it lacks rigorous alignment with the framework's stated goals. For these reasons, I do not believe this paper meets the standards for acceptance.

**Strengths:**

* **Precise Knowledge Unlearning Framework**: PROEDIT’s method incorporates a carefully designed approach to model parameter adjustment by defining unlearning, retention, and corruption scopes, allowing the model to selectively forget virus protein-related knowledge. This layered optimization helps to some extent in reducing the risk of affecting non-virus protein knowledge while also providing a degree of stability and adaptability to the model.

* **Comparison of Different Edit Methods**: The paper systematically examines various edit methods, including No Edit, Gradient Ascent, Joint Gradient Ascent and Descent, Random Labels, Gradient Ascent with KL Constraint, and the proposed PROEDIT. These comparisons provide clearer insights into the impact of each method on model performance and virus knowledge unlearning.

**Weaknesses:**

* **Unclear Justification of the Setting**: The paper fails to clearly explain why it is necessary to remove virus-related knowledge in protein language models, especially as this task is uniquely proposed by the authors themselves. It states, "This approach reduces ethical risks, specifically the ability of deep learning models to enhance the properties of viruses, while maintaining the model’s overall performance in designing normal, non-harmful proteins." However, the paper confuses the concept of "proteins present in viruses" with "harmful proteins." Not all "proteins present in viruses" are harmful. On the contrary, some viral proteins are beneficial for protein engineering; for example, the capsid protein in adeno-associated virus (AAV, https://en.wikipedia.org/wiki/Adeno-associated_virus) is widely used in gene therapy and vector design due to its low pathogenicity and efficient gene delivery capability. Such viral proteins play an important role in certain biomedical applications, so removing virus-related protein knowledge across the board may be inappropriate. Furthermore, in the paragraph starting with "The same concern has been raised in mutation effect prediction tasks," the paper provides no references to support why removing “virus protein” knowledge is necessary. Overall, the justification for this setting lacks persuasiveness.

* **Methodological Limitations**: The entire approach has fundamental limitations. According to the paper’s inference, protein engineering should aim to “design normal, non-harmful proteins.” However, using an unlearning approach cannot effectively achieve this goal. A more straightforward method would be to determine if a generated protein sequence is harmful (or, as the paper suggests, a “virus protein”). If a “virus protein” is designed, it could simply be rejected. In bioinformatics, this task is relatively simple. For instance, one can use BLAST or other homology search tools to compare the protein sequence with known “virus protein” databases, such as “virus protein” sequences in UniProt. High sequence similarity often suggests functional or origin-related associations. Alternatively, protein function databases like Pfam or InterPro can be used to identify functional domains. If the protein contains domains commonly found in “virus proteins,” such as reverse transcriptase or capsid proteins, it may be identified as a “virus protein.”

   Even if we insist on a machine learning approach, a binary classifier could be trained using “virus protein” data to predict whether a given protein is a “virus protein.” Using an unlearning framework forces the model to forget information about “virus proteins.” However, if the model does generate a “virus protein” (even with a lower probability), it will not recognize it as such. Thus, an additional predictor would still be needed to identify potentially harmful proteins, as outlined in the simpler approaches above.

* **Question Regarding Results on the AAV Dataset**: The PROEDIT results on the AAV dataset, shown in Table 2 and Figure 4a, appear questionable due to the negative Spearman correlation, which remains consistently negative throughout training—a common mistake when fine-tuning ESM models on supervised mutation effect datasets. Since Spearman correlation measures rank-order agreement, flipping the predictions' signs would make the correlation positive. In this context, considering the absolute value of Spearman correlation would be more meaningful, as it better reflects whether the model still retains “virus protein” information. The results suggest that PROEDIT, particularly at the 150M scale, may still retain virus-related knowledge and even outperform the pre-unlearning model, thereby casting doubt on the true effectiveness of the unlearning process.

**Questions:**

See **Question Regarding Results on the AAV Dataset** in weakness.

---

> ### Author Response · Authors · 2024-12-02
> **Clarifications and Future Directions**
>
> # Response to gH9B
>
> We appreciate your feedback on our work. We would like to address your questions and concerns as follows:
>
> > **Unclear Justification of the Setting**: The paper fails to clearly explain why it is necessary to remove virus-related knowledge in protein language models, especially as this task is uniquely proposed by the authors themselves. It states, "This approach reduces ethical risks, specifically the ability of deep learning models to enhance the properties of viruses, while maintaining the model’s overall performance in designing normal, non-harmful proteins." However, the paper confuses the concept of "proteins present in viruses" with "harmful proteins." Not all "proteins present in viruses" are harmful. On the contrary, some viral proteins are beneficial for protein engineering; for example, the capsid protein in adeno-associated virus (AAV, [https://en.wikipedia.org/wiki/Adeno-associated_virus](https://en.wikipedia.org/wiki/Adeno-associated_virus)) is widely used ...............
> >
>
> We understand that not all viral proteins are harmful, but we hold that removing harmful protein knowledge in large models is necessary for safety. The difficulty of obtaining a dataset of harmful viruses led us to focus on a widely-tested example, the AAV virus, for our experiments.
>
> > The entire approach has fundamental limitations. According to the paper’s inference, protein engineering should aim to “design normal, non-harmful proteins.” However, using an unlearning approach cannot effectively achieve this goal. A more straightforward method would be to determine if a generated protein sequence is harmful (or, as the paper suggests, a “virus protein”). If a “virus protein” is designed, it could simply be rejected. In bioinformatics, this task is relatively simple. For instance, one can use BLAST or other homology search tools to compare the protein sequence with known “virus protein” databases, such as “virus protein” sequences in UniProt. High sequence similarity often suggests functional or origin-related associations. Alternatively, protein function databases like Pfam or InterPro can be used to identify functional domains. If the protein contains domains commonly found in “virus proteins,” such as reverse transcriptase or capsid proteins, it may be identified as a “virus protein.” Even if we insist on a machine learning approach, a binary classifier could be trained using “virus protein” data to predict whether a given protein is a “virus protein.” Using an unlearning framework forces the model to forget information about “virus proteins.” However, if the model does generate a “virus protein” (even with a lower probability), it will not recognize it as such. Thus, an additional predictor would still be needed to identify potentially harmful proteins, as outlined in the simpler approaches above.
> >
>
> Training a classifier to determine if a generated protein is harmful would not be a practical solution. Unless the model is deployed as a server, users would be able to bypass any classification model. Once model weights are made public, developers lose control over the use of the model, making it impossible to guarantee safety. Moreover, this introduces another problem: how to ensure the safety of the harmful protein classifier itself. Therefore, our approach focuses on unlearning harmful protein knowledge at the model parameter level before the weights are made public. One key aim of our work is to draw attention to the safety of protein design models.
>
> > **Question Regarding Results on the AAV Dataset**: The PROEDIT results on the AAV dataset, shown in Table 2 and Figure 4a, appear questionable due to the negative Spearman correlation, which remains consistently negative throughout training—a common mistake when fine-tuning ESM models on supervised mutation effect datasets. Since Spearman correlation measures rank-order agreement, flipping the predictions' signs would make the correlation positive. In this context, considering the absolute value of Spearman correlation would be more meaningful, as it better reflects whether the model still retains “virus protein” information. The results suggest that PROEDIT, particularly at the 150M scale, may still retain virus-related knowledge and even outperform the pre-unlearning model, thereby casting doubt on the true effectiveness of the unlearning process.
> >
>
> We have honestly reported all results, and all fine-tuning code is consistent, so there are no errors in the reported results. We will include additional details about the training curves. While the training loss decreases, the validation loss continues to rise, suggesting that the model fails to generalize the knowledge of AAV, thereby demonstrating that the unlearning method can effectively disrupt the model's ability to generalize. However, we recognize that the choice of the AAV dataset may not have been ideal, and we will remove this dataset in future work and seek more appropriate datasets.

---

### Official Review · Reviewer_z71T · 2024-11-04

**Soundness:** 2
**Presentation:** 2
**Contribution:** 1
**Rating:** 5
**Confidence:** 3

**Summary:**

This paper presents a method called PROEDIT for “unlearning” virus-related knowledge in pre-trained protein language models (PLMs) to reduce ethical and biosafety risks in protein engineering tasks. This approach appears to be adapted from text LLM unlearning techniques. The authors choose a three-scope training scheme, including an unlearning scope, a retention scope, and a corruption scope, to selectively remove virus-specific data while retaining general capabilities on non-virus proteins.

**Strengths:**

1. The approach focuses on unlearning knowledge of the protein model, a specific verticle domain, which is novel.

2. The discussion of baselines in the experimental section appears sound.

**Weaknesses:**

1. I believe that prior work on unlearning knowledge in text-based LLMs, with similar designs, should be properly credited in the methods section.
The terms designed in the method section feel **similar** [1, 2] and (maybe thus) the design of the algorithm appears to lack explanation.
Unfortunately, I could not find any explicit citations or discussions on how you design (or analogy, or choose) the algorithm used, and I hope I missed something, as it would otherwise be**inappropriate**.

2. Compared to unlearning in text, the task of unlearning in the protein context feels somewhat artificial, with relatively limited significance.

3. Regarding the algorithm, is there a more principled way to select $D^{\text{sim}}$?

4. A minor issue: please keep the notation for  $x_M$  and $x_{/M}$  consistent.


[1] Chen J, Yang D. Unlearn what you want to forget: Efficient unlearning for llms[J]. arXiv preprint arXiv:2310.20150, 2023.

[2] Yao Y, Xu X, Liu Y. Large language model unlearning[J]. arXiv preprint arXiv:2310.10683, 2023.

**Questions:**

In the domain of safety alignment in text generation, there are methods to teach models to abstain. Is there any feasibility for this approach in protein modeling?

---

> ### Author Response · Authors · 2024-12-02
> **Response to concerns on unlearning methods in protein language models**
>
> We appreciate your feedback on our work. We would like to address your questions and concerns as follows:
>
> Weakness
>
> > (1) I believe that prior work on unlearning knowledge in text-based LLMs, with similar designs, should be properly credited in the methods section. The terms designed in the method section feel **similar** [1, 2] and (maybe thus) the design of the algorithm appears to lack explanation. Unfortunately, I could not find any explicit citations or discussions on how you design (or analogy, or choose) the algorithm used, and I hope I missed something, as it would otherwise be **inappropriate**.
> >
>
> Protein language models differ significantly from current large language models (LLMs) in natural language. Protein language models focus more on understanding task rather than generation. In mutant effect prediction tasks, autoregressive GPT-based models perform worse than BERT-based models focused on understanding. As a result, LLMs cannot be directly transferred to this domain. We will add references to the two papers in the related work section.
>
> > Compared to unlearning in text, the task of unlearning in the protein context feels somewhat artificial, with relatively limited significance.
> >
>
> This is indeed an important point. An example is ESM-3, a famous protein language model that achieves safety by removing viral data from the training set. However, even with this approach, the model still maintains some predictive capability for viral proteins. We hypothesize this may be due to the inclusion of proteins that are similar to, but not exactly viral, in the dataset. In such cases, methods like ours are needed to make the model safer and prevent harmful viral mutations.
>
> > Regarding the algorithm, is there a more principled way to select $D^{sim}$?
> >
>
> Sequence alignment algorithms such as BLAST or MMSeqs can also perform this task, but these methods can only search for sequences that are similar to a given protein, resulting in poor unlearning performance. These alignment methods lack the remote homology search capabilities that PLMs offer. Therefore, we use PLMs for this search.
>
> > A minor issue: please keep the notation for $x_M$ and $x_{/M}$ consisten
> >
>
> Thank you for the suggestion. We will correct this error.
>
> Questions
>
> > In the domain of safety alignment in text generation, there are methods to teach models to abstain. Is there any feasibility for this approach in protein modeling?
> >
>
> (1) This approach is certainly feasible for protein language models deployed as servers. For example, some structural prediction servers may refuse to process viral proteins as input, but this introduces a new challenge: how to determine whether a protein is virus.
>
> (2) For models with open weights, where users have access to the model's weights, this approach would no longer be effective.

---

### Meta-Review · Area_Chair_YbGU · 2024-12-17

**Metareview:**

This paper introduces a method (PROEDIT) to make a protein language model ``unlearn” virus-related info, aiming to reduce the model’s capability to enhance harmful viral mutations. It tries to keep good performance on non-virus proteins. That’s a neat safety angle, though somewhat speculative (in my view.

Regarding the rebuttal phase, the authors did respond, adding explanations about their approach and referencing related LLM-based unlearning methods when prompted by reviewers. However, the reviewers never fully converged on a positive stance. Their initial reservations—particularly about the ambiguous definition of ``harmfu” viral engineering—remained unsolved. There was also confusion on the choice of AAV data, which some reviewers found questionable as a benchmark.

Some common issues mentioned by multiple reviewers are: (1) The unclear or insufficient justification that removing all “virus protein” knowledge is necessary or effective at preventing harmful uses;
and (2) The questionable evaluation on AAV and other datasets, which leads to uncertain conclusions on the real effectiveness of the method. The paper’s main claims about safety benefits need more careful demonstration.

**Additional Comments On Reviewer Discussion:**

Regarding the rebuttal phase, the authors did respond, adding explanations about their approach and referencing related LLM-based unlearning methods when prompted by reviewers. However, the reviewers never fully converged on a positive stance. Their initial reservations—particularly about the ambiguous definition of ``harmfu” viral engineering—remained unsolved. There was also confusion on the choice of AAV data, which some reviewers found questionable as a benchmark.

---

### Decision · Program_Chairs · 2025-01-22

Reject